

# Increasing frequency and lengthening season of western disturbances is linked to increasing strength and delayed northward migration of the subtropical jet

Kieran M. R. Hunt[1,2]

[1]Department of Meteorology, University of Reading, Reading, UK
[2]National Centre for Atmospheric Sciences, University of Reading, Reading UK

**Correspondence:** Kieran M. R. Hunt (k.m.r.hunt@reading.ac.uk)

**Abstract.**

Western disturbances (WDs) are cyclonic storms that travel along the subtropical jet, bringing the majority of seasonal and extreme precipitation to mountainous South Asia in the winter months. They are a vital component of the region's water security. Although typically most common in the winter, WDs can also occur during the summer monsoon with catastrophic consequences. This happened earlier this year, leading to fatal floods across North India, including Delhi. Preceded by an unusually harsh winter season, questions are now being asked about how climate change is affecting WD frequency and intensity in both summer and winter seasons.

An analysis of 17 previous studies assessing trends in WD frequency revealed no consensus, at least in part because they quantified trends in different regions, seasons, and time periods. In this study, a more robust approach is used, quantifying trends in WD frequency and intensity by region and month, using a track catalogue derived from seventy years of ERA5 reanalysis data. Winter WDs have increased significantly over the Western and Central Himalaya and Hindu Kush in the last 70 years. This trend is attributed to a strengthening of the subtropical jet. The WD season has also significantly lengthened with WDs becoming far more common in May, June and July. For example, WDs have been twice as common in June in the last twenty years than during the previous fifty. This is attributed to delayed northward retreat of the subtropical jet, which historically has occurred before the onset of the summer monsoon. The most important implication is that the frequency of 'monsoonal' WDs is increasing significantly, and therefore, due to climate change, catastrophic events like the 2013 Uttarakhand floods and the 2023 North India floods are becoming much more frequent.

## 1 Introduction

Western disturbances (WDs) are upper-level troughs embedded in the subtropical westerly jet. Dynamical instabilities in the jet, often arising over the Mediterranean, grow baroclinically as they move eastwards, arriving as cyclonic winter storms over mountainous South Asia several days later (Singh, 1971). On average, six to seven strong WDs pass over the Indian subcontinent each month during winter (Rao and Srinivasan, 1969; Hunt et al., 2018a).



Here, they are responsible for the majority of winter mean and extreme precipitation (Hunt et al., 2018b, 2019a). WDs affect Iran, Afghanistan, Pakistan, and India, and occasionally extend their reach further east, to Nepal and Bangladesh (Pisharoty and Desai, 1956; Rao and Srinivasan, 1969). However, their impact is most strongly felt over north India and the Western Himalayas, where they cause coldwaves (Mooley, 1957; De et al., 2005), fog (Smith et al., 2022), avalanches (Ganju and Dimri, 2004), landslides (Hunt and Dimri, 2021), lightning (Unnikrishnan et al., 2021), and heavy precipitation through interaction with the orography (Singh and Agnihotri, 1977; Dimri and Chevuturi, 2014).

The importance of WDs lies in their contribution to winter precipitation, crucial for water security (Benn and Owen, 1998; Archer and Fowler, 2004; Thayyen and Gergan, 2010) and agriculture (Yadav et al., 2012). They therefore have direct and indirect economic impacts on India, Pakistan, and neighbouring countries. Extreme rainfall events associated with WDs can also cause severe flooding, especially near the start and end of the Indian summer monsoon (Dimri and Niyogi, 2012). These interactions with the monsoon can be dynamical or thermodynamical in nature (Hunt et al., 2021), but the increased convective instability and moisture supply provided by the monsoon can result in catastrophic flooding, for example resulting in over 6000 deaths in Uttarakhand in June 2013 (Kotal et al., 2014; Chevuturi and Dimri, 2016), and over 100 in North India in July 2023 (Associated Press, 2023).

Several methodologies have been used to quantify WD variability or trends over the past century: direct counting from various weather reports, feature-based tracking in reanalysis data, utilisation of filtered variance or clustering of appropriate meteorological fields in reanalyses, or inference from winter weather records in the Western Himalaya region.

Several studies, as depicted in blue in Figure 1, have employed weather reports to compile WD counts (Das et al., 2002; Shekhar et al., 2010; Ahmad and Sadiq, 2012; Kumar et al., 2015; Midhuna et al., 2020; Ahmed et al., 2022). These tend to report a decline in WD frequency, although this is not universal, and several are not statistically significant. Other recent research (Cannon et al., 2016; Hunt et al., 2018a; Nischal et al., 2021; Javed et al., 2022) has used feature-tracking algorithms applied to reanalysis data to generate WD variability records for trend analysis, none of which showed statistically significant trends. These are depicted in red in Figure 1. The third type of study, shown in green in Fig. 1, has typically used bandpassed upper-level geopotential variance as a proxy for WD frequency (Cannon et al., 2015; Madhura et al., 2015; Krishnan et al., 2018; Neal et al., 2020). These findings have revealed mixed results, including both significantly positive and negative trends in WD frequency. The use of geopotential variance as a proxy should be approached with caution, as it captures both WD frequency and intensity. Studies using this method tend to be more likely to report an increasing trend in WD intensity, unlike those focusing solely on WD frequency.

There is considerable disagreement among these studies regarding the trend in WD frequency over the past seventy years, with the trend being sensitive to the region of study, the seasonal definition, the definition of WD used, and even the chosen dataset period. Other factors, such as trends in the subtropical jet, decreasing static stability, and substantial interannual variability in jet behaviour further complicate trend interpretation (Abish et al., 2015; Yuval and Kaspi, 2020; Hunt and Zaz, 2023).

The aim of this paper is to resolve these disagreements using a more robust analysis method, relying on a database of WD tracks covering 1950 to the present day. We split the problem into four research questions:





**Figure 1.** Trends of WD activity since 1950 from the studies discussed in Sec. 1 that have available data. For each study, data are presented in the original units, with the stated season and impact region given. Those in blue derive their time series from observational records, such as IMD bulletins; those in red use tracking techniques applied to reanalysis data; those in green use variance-based methods applied to reanalysis data. For each time series, a black $+/-$ in the upper right corner indicates the trend is significantly different from zero at the 95% confidence level; a grey $+/-$ is used for the 50% confidence level.



1. Do trends in WD frequency vary spatially?

2. Do trends in WD frequency vary seasonally? If so, are WDs increasing in frequency in the pre-monsoon and monsoon seasons, extending their range beyond the winter months, as hypothesised by Valdiya (2020)?

3. Can differences in trends by explained by different definitions of WDs, namely using different intensity thresholds?

4. What are the causes of robust trends in WD frequency?

The reanalysis and WD track data used are described in Section 2, the results are presented in Section 3, and the results are summarised and their implications discussed in Section 4.

## 2 Data

### 2.1 ERA5

ERA5 is the fifth generation atmospheric reanalysis of global climate produced by the Copernicus Climate Change Service (C3S) at the European Centre for Medium-Range Weather Forecasts (Hersbach et al., 2020). Data from ERA5 (available from https://cds.climate.copernicus.eu/cdsapp#!/home) cover the entire globe on a 30 km grid and resolve the atmosphere on 137 levels from the ground up to 80 km in altitude. It covers from January 1940 until the present day at hourly frequency. At reduced spatial and temporal resolutions, ERA5 also includes uncertainty information for all variables. We use both pressure-level variables from ERA5, for both WD tracking (see Sec. 2.2) and to compute jet statistics.

### 2.2 Western disturbance track dataset

WDs are identified using the feature-tracking algorithm described in Hunt et al. (2018a). Relative vorticity is averaged across the 450-300 hPa layer, and then spectrally truncated to T42 ($\sim 300$ km at the equator) to remove high-frequency noise that hinders tracking. For each region of positive vorticity, the centroid is located and labelled as a candidate WD. These centroids are connected between timesteps using a nearest-neighbour algorithm, with a biased search to take into account the steering winds of the subtropical jet. Systems that do not on average travel eastward, last fewer than 48 hours, or do not pass through the box [20-42.5°N, 60-80°E] are rejected. Applied to ERA5, this gives over seventy years of track data (1950-2022). The method followed here is identical to Nischal et al. (2022), except the northern edge of the catching box is extended from 36.5°N to 42.5°N, to ensure that all WDs that potentially impact North India are included.

## 3 Results

We start by examining some mean statistics for WDs over the last seventy years (Fig. 2). The seasonal cycle of WD frequency (Fig. 2(a)) is extremely pronounced, with WDs being extremely common during the winter months (December to March), somewhat less common during the post-monsoon and pre-monsoon (October to November and April to May respectively),





**Figure 2.** The seasonal cycle of WDs, computed between 1950 and 2022. (a) Mean WD frequency for each month, computed as the number of WDs entering the region (50–80°E, 20–40°N). These are further stratified by overall intensity percentile, where intensity is defined as the maximum value of 350 hPa $\zeta$ that a WD reaches during its lifetime. (b) Map of average WD frequency from December to March, computed as the number of unique WD tracks that enter a given $2.5° \times 2.5°$ gridbox per month. The mean location of the subtropical jet is plotted in blue, computed for each longitude as the latitude during that season that has the highest mean 200 hPa zonal wind speed. (c), as (b) but for April and May. Smooth black lines indicate an orographic height of 2000 m.




and rather uncommon indeed during the summer monsoon (June to September). WDs are about eight times as frequent during January as in August. There is also a strong seasonality to their intensity. By stratifying the whole population into deciles of maximum vorticity (i.e., the peak intensity reached by a given WD during its lifetime), we see that the strongest WDs are proportionately far more common during the winter months. In contrast, during the monsoon, most WDs are substantially
weaker than the median. As a result, WDs reaching the top pentile of intensity are more than fifty time more common in January than August.

Alongside variations in frequency and intensity, WDs also exhibit marked seasonality in their spatial distributions. Track density during the winter months (Fig. 2(b)), when the subtropical jet is at its lowest latitude, is concentrated over the Western Himalaya and southern Karakoram, with high frequencies also present over the Central Himalaya and Hindu Kush mountains.
As the jet recedes poleward, eventually passing north of the Karakoram, during the pre-monsoon, track density also migrates northwards, lessening considerably over north India, but strengthening on the northern edge of the Tibetan Plateau and parts of the Karakoram. It also spreads out considerably, reflecting the greater variability in the subtropical jet behaviour after the winter season (Schiemann et al., 2009). Based on these maps, we assign a 'WD box' (50–80°E, 20–40°N) that captures the highest density of WD tracks over South Asia. This will be useful for statistical purposes later, as it also ensures that we only
use tracks that are likely to have an impact on Pakistan, north India, or the mountain ranges around the Western Himalaya.

Now that we are familiar with the mean and seasonal behaviour of WDs, let us investigate their trends. Since 1950, WD frequency over northern India has increased significantly, regardless of intensity (Fig. 3(a-b)). This includes the Western Himalaya, the Central Himalaya, and the Hindu Kush – winter WDs in these regions are becoming more frequent at a rate of about 20% per century. There is no significant signal in winter WD frequency over the Karakoram, however, and this is perhaps
surprising given that the theories explaining the Karakoram Anomaly[1] require a significant increase in snowfall over the region (de Kok et al., 2018).

During the pre-monsoon (Fig. 3(c)), this significant increasing trend persists across the Western Himalaya and Hindu Kush, but also extends over the Karakoram. Over north India, where WD frequencies are typically much lower during the pre-monsoon than the winter, trends of similar magnitude to those seen in the winter (Fig. 3(a)) indicate a much bigger relative
increase in WD frequency. This is potentially important for two reasons: firstly, as we have seen, these months are not typically well studied in the context of WD meteorology, and so trends here have thus far remained undiscovered; secondly, this increases the chance of WDs coming into contact with the summer monsoon, which can lead to catastrophic flooding (Ranalkar et al., 2016; Hunt et al., 2021; Kalshetti et al., 2022).

There is no signal in the trend of stronger WDs during the pre-monsoon (Fig. 3(d)), but this may because intense WDs are
uncommon during this time of year, and so remain dominated by interannual variability. In fact, the intensity of WDs at this time of year may anyway matter less than during the winter, as such systems (particularly in or after June) have an increasingly ready supply of moisture from the advancing monsoon. For example, the July 2023 floods over north India, which impacted

[1]This refers to the phenomenon that glaciers in the Karakoram are either growing or stable, counter to the expected impacts of global warming (Hewitt, 2005), and against the trend of glaciers in neighbouring ranges such as the Western Himalaya.





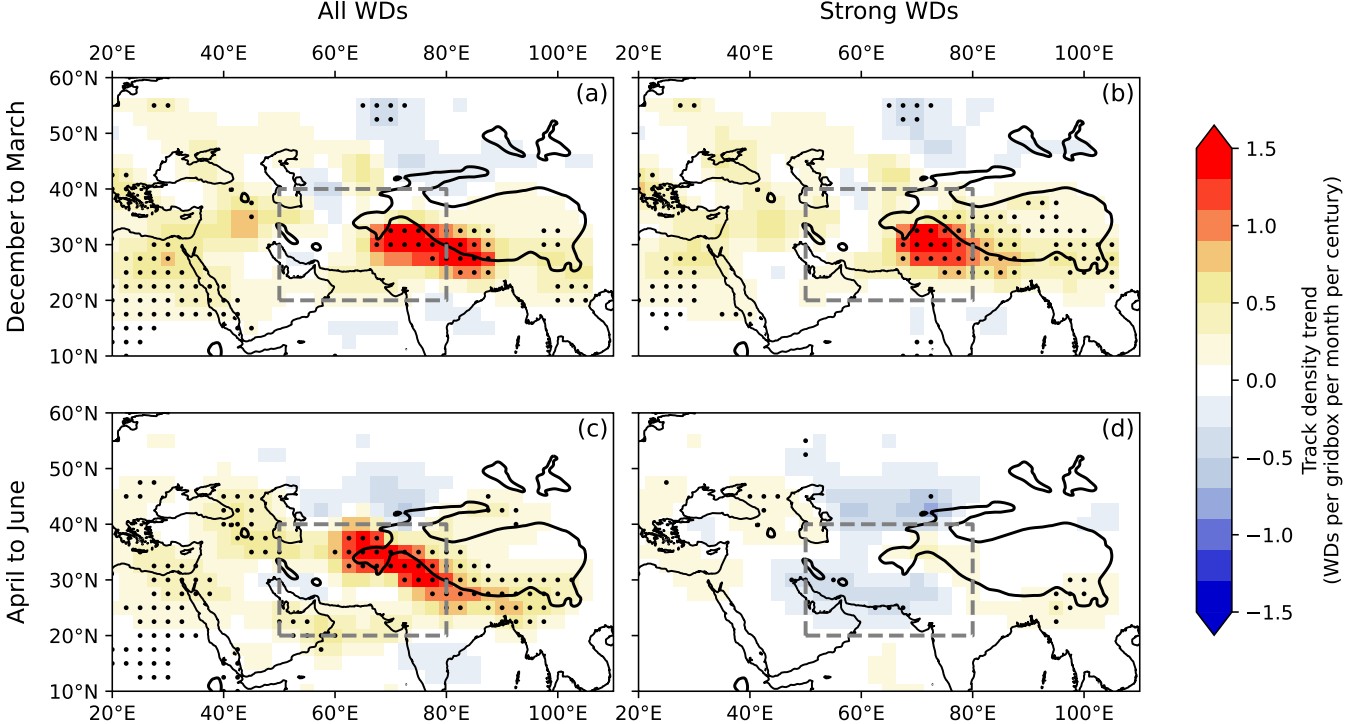

**Figure 3.** Trends in WD frequency from 1950 to 2022. The number of unique tracks passing through each $2.5° \times 2.5°$ gridbox per month is counted, aggregated by season, and the trend is computed. Two seasons are shown: (a) and (b) December to March, and (c) and (d) April to June. To determine whether trends are sensitive to WD detection thresholds, trends for "strong WDs", i.e., those whose peak 350 hPa $\zeta$ is in the top half of the population, are given in (b) and (d). Stippling indicates where trends have $p < 0.05$. The dashed grey box (50–80°E, 20–40°N) indicates the aggregation region for WDs used throughout this study.

a number of north Indian states, killing over a hundred people and resulting in extensive flooding in Delhi (Associated Press, 2023), were linked to the passage of a comparatively weak WD.

In general, these trends follow the mean track density (Fig. 2), implying that there has probably not been a significant shift in regions impacted by WDs. It is also unlikely that studies reporting negative trends in WD frequency can be explained purely through their choice of region.

     We now explore potential causes of these trends. It is known that both interannual and intraseasonal variability of WD frequency is strongly controlled by the latitude of the subtropical jet (Singh, 1971; Hunt et al., 2018a), so we can be fairly

confident in a hypothesis that explains trends in WD frequency in terms of trends in upper-level zonal winds. Indeed, there has been a significant increase in the wind speed of the upper-level westerlies over northern India and the Himalaya over the last seventy years (Fig. 4).

     Changes in the subtropical jet are often linked to hemispheric-scale changes in the upper-level meridional temperature gradient (e.g. Stendel et al., 2021), but it is clear from the maximum over the Tibetan Plateau that there is also a locally-forced





**Figure 4.** Trends in 200 hPa seasonal mean zonal wind from 1950 to 2022. (a) December to March and (b) April to June. Stippling indicates where trends have $p < 0.05$. The dashed grey box (50–80°E, 20–40°N) indicates the aggregation region for WDs used throughout this study. The histograms on the right of each subplot show the latitudinal distribution of the jet axis for the respective season, computed as the latitude at which the monthly mean 200 hPa zonal wind reaches its peak when averaged across 50–80°E.





change here as well. The sign of the trend, and its local maximum here, is consistent with earlier studies that have investigated the relationship between the winter subtropical jet and climate change over a similar period (e.g. Pena-Ortiz et al., 2013). This local response is crucial, because it amplifies the effect of climate change on WDs, not only through modifying the jet, but by making the local environment more unstable (Krishnan et al., 2018). This local anomaly occurs directly as a result of the Tibetan Plateau having warmed more quickly than neighbouring regions and acting as an elevated heat source in the
mid-troposphere (Wang et al., 2008).

The increasing trend in 200 hPa $u$ is present in both the winter and pre-monsoon seasons, and could thus explain increasing WD frequency in both seasons. However, larger mean $u$ over north India could mean either that the jet is getting stronger, or simply that it is *more likely* to be in this position. The histograms on the right of Fig. 4, showing the distribution of jet latitude for each season resolves this. In the winter months, the jet latitude distribution has a strong peak at 28°N, with very
little variance. This coincides with the region of increasing $u$, implying that during the winter, the jet has strengthened over recent decades. In the pre-monsoon months, jet latitude varies considerably, and is usually found between 30°N and 40°N. This is largely to the north (though with some overlap) of the region of significantly increasing $u$, implying that in pre-monsoon, the jet is more likely to be located further south in recent decades than previously. Disentangling the relative effects of these contributions is vital for a full explanation of WD trends and requires more in depth statistics, as we will see.

Part of the problem is that WDs respond to changes in the subtropical jet in different ways in different seasons. The most obvious example is the relationship between jet latitude and WD frequency. During the summer monsoon, when the jet is far north, we would expect a negative correlation between jet latitude and WD frequency, as anomalous southward excursions of the jet are more likely to bring WDs to north India. Conversely, in the winter months, anomalous southward excursions of the jet move it away from the baroclinic environment of the Himalaya, reducing WD frequency and intensity (Baudouin et al.,
150    2021).

For this reason, we now investigate the relationships between jet strength and latitude and WD frequency and intensity for each month (Fig. 5), allowing appropriate partitioning of effects into the winter and pre-monsoon months. The latitude of the jet core over South Asia has increased (i.e., moved northward) in all months from November to March, with a signficant trend in all months except February. Conversely, in the pre-monsoon and summer monsoon, the jet has tended to a more southward
position over the last seventy years, although only significantly so in June and August (Fig. 5(a)). This signal is particularly important during June, as typically the jet starts to migrate northwards during the onset of the summer monsoon (Schiemann et al., 2009). The negative correlation here implies that migration has become increasingly delayed in recent years. Jet core strength, on the other hand, has seen a significant increase in each month from October to May (Fig. 5(b)). This is in agreement with Fig. 4.

Within the WD box, monthly WD frequency has increased in every month over the last seventy years (Fig. 5(c)). However, the trends are only statistically significant in January and from April to July (i.e., through the whole pre-monsoon and monsoon onset). This, alongside substantial interdecadal variability, may explain why previous studies have disagreed on their assessments of WD frequency trends. We note, however, (not shown) that moving to a 1979 start, consistent with many studies (e.g., those based on older reanalyses), makes little difference to the sign or significance of the trends presented here. For comparison,





**Figure 5.** Trends in selected WD and subtropical jet monthly statistics. Jet statistics are computed as an average between $50°$E and $80°$E. WD statistics are computed over the WD box used throughout ($50–80°$E, $20–40°$N). At the end of each row, a '+' indicates a positive trend and a '−' indicates a negative trend. These are coloured black and grey respectively for $p < 0.05$ and $p < 0.5$. (a) latitude of the jet core, measured as the latitude where the zonal average of 200 hPa $u$ is maximum. (b) strength of the jet core, taken as the maximum of the zonal average of 200 hPa $u$ (i.e., the zonal wind speed at the mean core location). (c) the number of unique WD tracks that enter the WD box within a given month. (d) the mean intensity of those WDs (again measured only within the box), taken as the 350 hPa $\zeta$ measured at the WD centre. (e) Pearson correlation coefficients between selected pairs of these statistics for each month. The grey band indicates where $p > 0.05$. For (a)-(d), the annual timeseries have been smoothed with a 2-year Gaussian low-pass filter for clarity. However, only the original timeseries are used for computing statistics.



mean WD intensity has significantly declined in April and May, although the relative magnitude of this decline is small in the context of the population mean. Insignificant positive trends dominate the winter months. The increases in WD frequency are both particularly prominent and appear to be recent phenomena: in May, WDs have historically occurred at the rate of about 10 per month, and this has risen to about 14 in the last twenty years; in June, they have risen from a historical rate of about 4 per month to nearly 8 per month in the last ten years.

To help us causally connect the behaviour of the subtropical jet to WDs, we can also compute the annual cycles of correlations between selected pairs of these statistics (Fig. 5(e)). Firstly, jet characteristics are typically not a good predictor of monthly mean WD intensity, with which both jet strength and latitude are typically only weakly correlated. The exception is in the height of the winter (December and January) when WD intensity is significantly positively correlated with jet latitude. As we have already discussed, southward deviations of the jet from its mean position (about 28–30°N during these months) take WDs

away from both mountain- and circulation-driven regions of baroclinic instability.

The significant increase in January WD frequency can be attributed to increased subtropical jet strength over the last seventy years, consistent with Fig. 4(a). A stronger jet offers deeper vertical wind shear and hence a greater source of baroclinic instability for WDs to form and grow Sankar and Babu (2021); Nischal et al. (2023). In contrast, as we have seen, the increase in WD intensity in January is controlled more by jet latitude. The correlation between January jet latitude and strength is

significantly positive, but not large enough that either can be dismissed as a confounding factor.

WDs have also increased significantly in every month from April to July, covering much of the pre-monsoon and monsoon seasons. In April, this appears to have been driven by the sharp increase in jet strength, whereas in June and July it appears to be driven by a more equatorward positioning of the jet. We can probably rule out jet strength as a confounding factor in these months because the only known mechanism through which a weaker jet produces more WDs is through a latitudinal shift in its

position; otherwise decreased baroclinicity would cause a decline in frequency. May appears to represent a transitional period between these two effects.

In summary, WD frequency has significantly increased over the last seventy years in January and April through July. In January and April, we attribute this to increasing jet strength; in June and July we attribute this to decreasing jet latitude – in other words, the jet is more likely to remain over northern India and Pakistan as the monsoon develops, rather than receding

northwards as it has historically done so.

## 4   Conclusion

Western disturbances (WDs) are synoptic-scale cyclonic circulations that propagate westward along the subtropical jet, bringing heavy precipitation to Pakistan and northern India, particularly to the mountainous regions therein. They are a vital component of the region's water security, recharging glaciers and the snowpack as well as providing rainfall for irrigation at lower

elevations. Although they are most common in winter (December to March), they can occur, albeit unusually, at any time of year. In rare cases, strong WDs can interact with the summer monsoon, producing catastrophically heavy rainfall.





Given this, it is important to quantify trends in WD frequency and intensity, so that adequate preparations can be made. However, previous studies have largely disagreed on the sign and significance of WD frequency trends. In this study, we took a more robust approach, breaking down the trends both spatially and seasonally, using seventy years of reanalysis data.

Key results are as follows:

- WDs, as measured by their 350 hPa relative vorticity, are much stronger in winter months. WDs whose peak intensity exceeds the 75th percentile virtually never occur between June and September.

- WD track density varies considerably between the winter and pre-monsoon (here taken as April to June). During the winter, tracks predominantly impact the Hindu Kush and Western Himalaya, and can often persist long enough to reach

the Central and Eastern Himalaya. During the pre-monsoon, track density is more spread, with many more impacting the Karakoram, and relatively few impacting the Central or Eastern Himalaya.

- WDs have increased significantly over the Western Himalaya and Hindu Kush in winter, and significantly over these regions and the Karakoram in the pre-monsoon and early monsoon. As these approximately follow mean track density, there has been no marked shift in seasonal track density, nor do any regions show a significant decline in WD frequency.

- Upper-level zonal wind has increased significantly over north India and Pakistan in both the winter and pre-monsoon seasons. This implies the subtropical jet has increased in intensity in the winter, and is more frequently situated over the region during the pre-monsoon and early monsoon. Trend maxima over the Tibetan Plateau imply a strong local contribution to the forcing in circulation.

- Over a box incorporating most of their population (50–80°E, 20–40°N), WDs have significantly increased in frequency

in January, April, May, June, and July. Increases in May and June over the last twenty years have been particularly notable – for example, WDs in June are now twice as frequent as they were for much of the twentieth century. This implies a considerable lengthening of the WD season, and therefore a significantly increased risk in WDs interacting with the summer monsoon.

- We attributed January and April frequency increases to a strengthening of the subtropical jet, and the June and July

increases to a delayed northward retreat of the jet.

- Trends in WD intensity are statistically significant only in April and May, where they have declined slightly. We were not able to link these to changes in jet behaviour, and so this remains an open question.

This changing seasonality of WD frequency does not wholly explain the disagreement between earlier studies in establishing the sign and significance of historical WD frequency trends. Regional variations in these trends do not appear to explain why

some studies have reported negative trends in WD frequency, but do partially explain the mix of no, insignificantly positive, and significantly positive trends reported in previous studies, as the region of significantly positive trend in winter WD frequency reported here covers only the Hindu Kush and Western and Central Himalaya, but not the Karakoram, a region often included



in such studies. There is has been no significant spatial shift in WD track density in the last seventy years, and so it is unlikely that this is the cause of previous disagreement.

One possible resolution, which has not been explored here, is the role of interdecadal variability. For example, WD frequency is known to be correlated with the North Atlantic Oscillation (Yadav et al., 2009; Filippi et al., 2014; Hunt and Zaz, 2023), and correlating monthly mean SSTs with WD frequency also suggests strong modulation by the Pacific Decadal Oscillation (not shown). Even if there is a long-term trend, which the analysis presented here supports, shorter study periods sampling different parts of the interdecadal variability would lead to conflicting answers.

Aside from this, an important shortcoming of this paper is the dependence on a single reanalysis. Not only should trends drawn from reanalyses be interpreted cautiously due to changes in observing systems (Thorne and Vose, 2010), but the reader should be aware that tracked datasets drawn from different reanalyses can differ considerably (e.g., for extratropical cyclones Hodges et al., 2011). It is also worth noting that the results presented here are somewhat at odds with those previously presented by the author (Hunt et al., 2019b), in which a similar method applied to CMIP5 models indicated a significant decline in WD

frequency and intensity, attributed to weakening baroclinicity. A more thorough review using CMIP6 model output is required.

*Data availability.*    ERA5 data were downloaded from https://cds.climate.copernicus.eu/cdsapp#!/dataset/reanalysis-era5-pressure-levels?tab= overview, where they are freely available. The western disturbance track data are freely available on Zenodo, at https://doi.org/10.5281/ zenodo.8208018.

*Competing interests.*    The author declares no conflict of interest.

*Acknowledgements.*    KMRH is supported by a NERC Independent Research Fellowship (MITRE; NE/W007924/1). KMRH also wishes to thank JP Baudouin, a discussion with whom inspired this paper.



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
