# Peer review of "Increasing frequency and lengthening season of western disturbances is linked to increasing strength and delayed northward migration of the subtropical jet"

_EGUsphere, 2023_

## Author Response (AR1)

**Reviewer 1**

Review comments on "Increasing frequency and lengthening season of western disturbances is linked to increasing strength and delayed northward migration of the subtropical jet" by  Kieran M. R. Hunt
I thank Prof Dimri for his positive evaluation of this paper. I respond to his comments below, point-by-point, in red. Changes to the manuscript are detailed in blue.

"mountainous South Asia" should be "Hindukush Himalayas" ONLY. Else mountainous South Asia also includes Khasi Jantia Hills in far east and in Burma too where WDs do not reach.
I agree, I'll make this change.

Change ".....WDs can also occur during the summer monsoon with catastrophic consequence..."
To '....WDs can also occur during the summer as well interacting with monsoon leading to catastrophic 5 consequence
I will change this line to "WDs can also interact with the summer monsoon leading to catastrophic consequences"

Change "...WD season has also significantly lengthened with WDs becoming far more common in May, June and July." to "....WD season has also significantly lengthened with WDs freaking in May, June and July"
"Freaking" isn't a word, but I will adjust the sentence to reflect the unusual nature of the shift: "The WD season has also significantly lengthened with WDs becoming far more common in May, June and July; months where they were previously rare."

Pls see following as well.
Western Disturbances: A review. A. P. Dimri, D. Niyogi, A. P. Barros, J. Ridley, U. C. Mohanty, T. Yasunari, D. R. Sikka. Reviews of GeophysicsVolume 53, Issue 2 p. 225-246. https://doi.org/10.1002/2014RG000460
Ok, I'll add this reference where relevant in the introduction.

This is exceptionally good paper and work. Kieran has proposed a new dimension on SWJ dynamics leading to determine the WDs'.

I strongly recommend this paper to accept.

Dimri

**Reviewer 2**

The study discusses trends in mid-latitude upper tropospheric disturbances reaching Pakistan and NW India, also known as Western Disturbances (WD). As the author pointed out, the review of the literature is quite inconclusive, and a new study on the topic is welcome. The study uses a well-proofed detection technic that has been used to answer many scientific questions regarding WD in the past years. The observational dataset (ERA5 reanalysis) used for the analysis is also one of the best available for that kind of study, and its recent extension to dates prior to 1979 also enables potentially more robust trend analyses. In addition to these sound context review and framework, the study also brings clear and exciting new results on WD characteristic trends depending on the seasons. Yet, I would suggest clarifying the following points, in order to further improve the quality of the paper: I thank Dr Baudouin for his positive and critical evaluation of this paper. I respond to his comments below, point-by-point, in red. Changes to the manuscript are detailed in blue.

- I am wondering how robust the results are with respect to the dataset. The extension of ERA5 before 1979 is relatively new. It could be suffering from the assimilation of far fewer observations than the rest of the time period, and presenting spurious trends. In particular, a drop in the number of upper troposphere observations could be particularly detrimental to the analysis. For example, could it be possible that the fewer assimilated data in the early part of the reanalysis results in a blurring of the field used for WD detection, and therefore decrease the detection rate / intensity of the WDs? Has any study yet tried to validate upper-level tropospheric fields in ERA5? Could observations from radiosondes be used to validate some results of the study? (e.g. from https://www.ncei.noaa.gov/products/weather-balloon/integrated-global-radiosonde-archive)
This is a reasonable concern, and it's one of the reasons that I truncate at 1950 rather than going all the way back to 1940 (when ERA5 starts). We can't really use radiosonde measurements to verify upper-tropospheric fields in ERA5 because they are ingested in the product itself. However, we can look at the trends themselves, e.g. in Figure 5. Where the trends are significantly positive or negative, you can see that behaviour is largely consistent across the whole length of the dataset, but the magnitude of the trend tends to increase in the last few decades (as I have noted in the text in a few places). To verify this, I replot Figure 5 below, removing 1950-1978 from the analysis. You can see that almost all signs remain the same, as do most of the significances.

[Figure]

- I am concerned that the paper could be seen as "yet another study with a different conclusion on WD activity trend". Could the author further analyse the differences between the new analysis compared to the previous studies? What should the reader do with all these seemingly contradicting studies? Even if the author cannot explain or suggest the reasons behind the differences with all the studies in Fig.1, particular attention should be put on Hunt et al. 2018 and Nischal et al. 2021. Those two studies are based on the very same algorithm and dataset as the current paper, and yet present respectively no trend, or a weak positive one. Are the results from this detection algorithm actually robust enough?
This is a good question, and it is important to resolve this apparent discrepancy. Firstly, there is a small mistake in the methodology, which I have now corrected. The process isn't completely identical to these earlier papers because I use a T42

truncation rather than a T63 truncation to remove noise from the relative vorticity field (I will, of course, correct this in the revised methodology). In reality, this makes very little difference to the dataset, but does make the algorithm better at capturing broad-trough WDs with multiple vorticity maxima. In the figure below, I show the DJFM WD frequency trend for selected boxes, including the one used in this study and the slightly smaller one used in the Hunt/Nischal studies. You can see that the positive trend is robust across the choice of box, although its statistical significance varies as a function of box size (consistent with Figure 3 in the paper, and due to how much area outside the Himalayas/Karakoram, where the trend is very strong, is included). Therefore, for the box used in Javed et al., focused only on the Western Himalaya, the trend is very positive with high significance. For the larger boxes used in this study and the previous Hunt/Nischal studies, the significance hovers around p=0.05, which explains why when different datasets (e.g. ERA-Interim) or different truncations (e.g. T63) are used, the trend can lose significance.

The corrected methodology now reads: "The method followed here is almost identical to Nischal et al (2022), except the northern edge of the catching box is extended from 36.5°N to 42.5°N, to ensure that all WDs that potentially impact North India are included; and the spectral truncation is at T42, rather than T63, as this better captures broad-trough WDs with multiple vorticity maxima.".

[Figure]

WD frequency trends for different boxes (DJFM)

- I haven't really understood the purpose/design/discussion of the data from the right panel in figure 4, and panel a/b in figure 5, regarding jet characteristics. The author rightly notices that changes in the mean wind could be related to either jet speed or jet variation around its main position (L. 137-138). However, to distinguish between the two, the use of daily data (or higher frequency) would be needed. In fact, the algorithm from Schiemann et al. 2009 could be used to investigate trends in the entire distribution of jet latitudinal position and maximum intensity and would potentially give better insights into the jet behaviour.

This is a good point, and it is plausible that the two effects (jet strength and jet latitude) could be confounded by using monthly means. I have included an edited version of Figure 4, given below. This adds on a climatological jet axis for each season, computed using daily U200, as well updating the histograms on the right to use daily data. These histograms show the mean latitude of the jet axis averaged between 50°-80°E, but now include histograms from the beginning (1950–1970; blue) and end (2000–2020; orange) of the study period. These additions add more evidence to the original arguments: the Dec-Mar jet is getting stronger (no significant shift in latitude and the jet axis runs right through the region of increasing U200) and the Apr-Jun jet is shifting southward (significant change in distribution and increasing U200 to the south of the jet axis).

[Figure]

- Finally, there was little attempt to discuss the reason for the jet shift, and whether one should expect an amplification of this phenomenon in a warming world. The argument on TP high warming rate would rather explain an earlier jet shift to the North, rather than a delay (cf. Krishnan et al. 2018). And despite citing several studies that have investigated jet trends, I don't see in the paper a comparison between their results to the ones presented here. In addition, from Figure 4, I see that a dipole of wind speed anomaly is evident, with the negative anomaly over the tropics being the strongest. I am wondering whether this suggests a tropical origin to the trend, e.g. shift of the Walker circulation, stronger subsidence of the Hadley cell branch over N India, increased convection over Indonesia?

This is a good question. I plot below the trend computed from 1950-2022 of DJFM (left) and AMJ (right) 2-m temperature (top) and 250 hPa temperature (bottom). The reviewer is correct that the anomalous near-surface heating of the TP sets the temperature gradient in the wrong direction.

[Figure]

A full investigation of the cause of the jet changes is beyond the scope of the paper, but I agree that a better literature summary could be conducted. There seem to be three complementary theories linking a stronger winter subtropical jet to climate change:
1) Walker circulation modulation, as proposed by the reviewer above (local effect; Liu et al 2021)
2) Aerosols (local effect; Abish et al 2014; Chemke and Dagan 2018)
3) Increased deep convection in the tropics increases the upper-tropospheric meridional temperature gradient (global effect, Woolings et al 2023; Menzel et al 2019)

I have removed the incorrect statement and replaced with the following discussion:
"This locally-amplified increase in subtropical jet strength could be caused by shifts in the Walker circulation (Liu et al., 2021) or trends in aerosol loading over north India (Abish et al., 2015; Chemke and Dagan, 2018), whereas the global-scale increase in the meridional gradient of tropical upper-tropospheric temperature is likely due to increased convection in the tropics (Menzel et al., 2019; Woollings et al., 2023)."

Abish, B., Joseph, P. V., & Johannessen, O. M. (2015). Climate change in the subtropical jetstream during 1950–2009. *Advances in Atmospheric Sciences*, 32, 140-148.

Chemke, R., & Dagan, G. (2018). The effects of the spatial distribution of direct anthropogenic aerosols radiative forcing on atmospheric circulation. *Journal of Climate*, 31(17), 7129-7145.

Liu, X., Grise, K. M., Schmidt, D. F., & Davis, R. E. (2021). Regional characteristics of variability in the Northern Hemisphere wintertime polar front jet and subtropical jet in observations and CMIP6 models. *Journal of Geophysical Research: Atmospheres*, 126(22), e2021JD034876.

Menzel, M. E., Waugh, D., & Grise, K. (2019). Disconnect between Hadley cell and subtropical jet variability and response to increased CO2. *Geophysical Research Letters*, 46(12), 7045-7053.

Woollings, T., Drouard, M., O'Reilly, C. H., Sexton, D. M., & McSweeney, C. (2023). Trends in the atmospheric jet streams are emerging in observations and could be linked to tropical warming. *Communications Earth & Environment*, 4(1), 125.

I also have a few minor comments and questions along the text, including a few typos:

L.5 "This happened ...": This sentence sounds off in the abstract, it's oddly contextual, as summer WDs are not the main point of the paper. It should rather go in the introduction.
I have changed this part of the opening paragraph to: "Although typically most common in the winter, WDs can also interact with the summer monsoon leading to catastrophic consequences. These seem to be happening more frequently, and along with increasingly harsh winter seasons, questions are now being asked about how climate change is affecting WD frequency and intensity in both summer and winter seasons."

L. 53: "such as trends ...": these trends are rather potential causes of WD change rather than sources of confusion, or I am missing the author's point.
The point is that there are many factors that could influence local trends in WD frequency, and depending on how and where WD frequency is measured, may lead to conflicting signals. For example, a latitudinal shift in the jet may change WD frequency over a region (which may in turn be the focus of one of the trend studies), but not "overall" WD frequency.

Fig 1. and 5: I didn't notice before reading the legend that some signs were grey, and others black. Maybe increase the difference?
Yes, I've done this.

L. 61: "Can differences in trends by explained" -> Can differences in trends be explained
Thanks, I've fixed this.

L.61. "using different intensity thresholds": despite this being one of the main questions of the author, this is only very quickly investigated in Figure 3, and the text does away with the difference as fast. This question should either be further explored or not be one of the main questions.
Yes, this is fair. I do also examine intensity as one of the key parameters in Figure 5, where I show there has been no significant trend in WD intensity in any month except for April and May. The relevance of this should be discussed in the text, where I have now added: "For comparison, mean WD intensity has significantly declined in April and May, although the relative magnitude of this decline is small in the context of the population mean. Insignificant positive trends dominate the winter months. These results are consistent with Fig. 3, which showed that the frequency of WDs significantly increased in both winter and spring, but that strong WDs increased only in winter (with no significant spring trend). Taken together, this implies that there is no significant trend in the intensity of winter WDs and thus disagreements between studies confined to those months (i.e., December to March) cannot be explained by different choices of WD intensity threshold.".

L. 79 "[20-42.5°N, 60-80°E]": does the size of the box impact the robustness of the trends (for example, wrt Nischal et al 2022 who used a smaller box). Also, this removes most of the mid-latitude disturbances that pass over Central Asia. It seems from Figure 3c that those are rarer. Does this characterise a southern shift of the spring storm track, alongside the jet?
See my response to your major comment #2 for a discussion on trend robustness as a function of box size/location. Storm tracks over Central Asia are not of particular interest as these do not impact the Himalaya or South Asia, but you are correct in that there is an implied equatorward shift in the storm track. I have mentioned this briefly in the revised text: "There is a weak but significant decline in the number of systems passing over central Asia, i.e., that are deflected north of the Tibetan Plateau, during the winter. These are already rare, and have little impact on the weather of the Western Himalayas, so we leave a more detailed investigation for later research.".

L. 79 "1950-2022": any reason for not using the 1940-1949 time period of ERA5?
Yes, upper air soundings across central Asia were sparse before 1950 (see e.g., appendix tables of Lanzante et al., 2003), and so estimates of jet wind speed or disturbances embedded therein come with much greater uncertainty. For that

reason, I only carry out WD tracking (and hence all following analysis) from 1950 onwards. I mention this in the revised methodology: "Applied to ERA5, this gives over seventy years of track data (1950--2022). Although the ERA5 dataset starts in 1940, we only use data from 1950 onwards because upper air soundings over Central and South Asia were very sparse before this date (LAnzante et al., 2003), leading to large uncertainties in the reanalysis fields in our study region."

Lanzante, J. R., Klein, S. A., & Seidel, D. J. (2003). Temporal homogenization of monthly radiosonde temperature data. Part I: Methodology. *Journal of Climate*, 16(2), 224-240.

Fig 2 (legend) : "for April and May": the panel c says "April to June"
Thank you – I have corrected the figure caption.

Still Fig.2: Why does it seem that the track density is higher in c) than in b), when WD frequency is higher in the winter period overall?
Good spot – there was an error in the normalisation calculation which I have now fixed. The updated figure is below.

[Figure]

L. 97: "reflecting the greater variability in the subtropical jet behaviour after the winter season": More specifically in April/May and October/November, as the jet is instead more stable in summer than in winter.

I agree with the proposed change in wording and have adopted it. However, I disagree that the jet is more stable in summer than in winter, as you can see from the distribution of jet latitudes for the two seasons in the figure below.

[Figure]

**(a) December to March**

**(b) April to June**

L. 98: "WD box": why not take the same box as for the detection algorithm? That would be more consistent.
It would be consistent, but perhaps not as useful. The detection box is designed to capture any systems coming into the region, regardless of impact, that could reasonably be called a western disturbance. The assigned WD box is chosen to select the region of highest WD track density – this increases the signal-to-noise ratio of later statistical analysis and also ensures that systems that pass to the north of the Karakoram and thus have little impact (due to scarce moisture availability) are not included.

L. 102: "significantly": it should be specified at least once in the text that this corresponds to a statistical test.
Agreed, I have made this change.

L. 105 "significant" : substantial (because it's not about the statistical test)
I agree – have made this change.

L. 105 "snowfall": but precipitation is not really co-located with the WD centre.
Trends in precipitation in relation to WD would be a whole new paper I guess.
This is true, of course, but if there are more WDs passing over the region, it is
reasonable to assume they should be associated with increased seasonal snowfall.

L. 121: "not been a significant shift in regions impacted by WDs": This is very
interesting, but I am a bit puzzled. Since the jet has shifted, why hasn't the WD track
too? Is it because of some intensity threshold? For example, the meanders in the jet,
when it is south enough, do not intensify enough (because of the lack of relief
interaction) to be detected by the algorithm? Do these immature WDs still have an
impact on the ground though?
This is a good question.  The track doesn't shift much compared to overall WD
distribution, but perhaps more crucially, WD formation is quite local, so a jet shift
just results in fewer WDs rather than necessarily a big shift in their location. I have
explained this in the revised manuscript: "As we will see, there have been nonzero
trends in jet latitude over the study period, but as WDs often spin up locally (i.e.,
requiring the baroclinic environment provided by the orography), this seems to
result in a change in frequency more than a change in location."

L. 139 "resolve this": Besides my disagreeing (See main point 3), this seems to be
contradicted a few lines after: "requires more in-depth statistics" (L.144)
In response to your main point 3, I have replotted Figure 4 using daily statistics
(which you can see in that response). The new figure provides much better evidence
supporting the winter strengthening/spring shifting hypothesis, and so I have
removed the "requires more in-depth statistics" comment.

L. 140 "This coincides with the region of increasing u": Though the main anomaly is
located to the North of the distribution's mode.
This is true, although the jet is quite wide, so much of it covers this region anyway. I
will edit the text to reflect this subtlety: "In the winter months, the jet latitude
distribution has a strong peak at 28°N, with little variance. This largely coincides
with the region of increasing $u$, although the trend maximum is located slightly to
the north of the jet axis, implying that during the winter, the jet has strengthened
over recent decades. The distribution of jet latitudes has also shifted slightly
northwards, as we see from comparing the 1950–1970 and 2000–2020
distributions."

L. 145: "in different seasons" -> depending on the season (?)
I have made this change.

L. 156: "important" -> impactful, consequential?
I have made this change.

L.156: "during June, as typically the jet starts to migrate northwards ...": It starts to move in April / May. It's pretty much already established North of the TP in June (Schiemann et al 2009)

This is a fair point, although it can still linger around south of the TP well into June (as happened this year, for example). I have corrected the wording to make it more accurate: "This signal is particularly consequential during May and June, as typically the jet starts to migrate northwards before or during the onset of the summer monsoon (Schiemann et al., 2009). The negative trends here imply that migration has become increasingly delayed in recent years."

L. 157: "The negative correlation": what correlation? isn't it rather the negative trend (in intensity)?

This refers to the trend in jet latitude – I have corrected this.

Fig. 5c/d: cutting the WD detection at 40°N removes the possibility to see how the mid-latitude disturbances over Central Asia behave.

I agree, but (1) they are not the focus of this paper, (2) they do not bring substantial impacts to the region of interest (Hindu Kush, Karakoram, and Himalaya), and (3) as has already been pointed out, they slightly decrease, so including them in the general population would weaken the trend signals, as has been a problem with prior studies.

Fig.5e: I suggest having the same colour for the two "WD *, jet strength", and "WD *, jet latitude" respectively. Also, do WD frequency and intensity correlate?

I give a potential revision for Figure 5e below, although I don't think it is quite as clear as the original. WD frequency and intensity do not correlate much on a monthly basis (although the correlation is strong over the whole population of course!)

[Figure]

L. 167 (and later) "historically": it's vague, and it suggests that WD frequency normally doesn't vary, which is untrue since at least decadal variability is presented. Which time period do you consider here?

This is used in contrast to claims made about the last twenty years, so I have corrected these instances to read "prior to 2000".

L. 175 "take WDs away": maybe clarify that the WD would still be detected, but wouldn't intensify as much?
Sure, I have now added "Such WDs would still be detected but would not intensify as much as their counterfactuals." here.

L. 177 "A stronger jet offers deeper vertical wind shear". The vertical structure of the wind change is not investigated here, a stronger jet could also only increase the upper troposphere wind shear.
Although it is a reasonable assumption that a stronger jet should be associated with stronger vertical wind shear, there are cases where this may not be true. To confirm this claim, I have included a figure below which shows the trend in the vertical shear of zonal wind (200 hPa minus 500 hPa) for DJFM over the period 1950 to 2020. The mean jet runs through a region of increased vertical wind shear.

[Figure]

L. 178: "Sankar and Babu (2021); Nischal et al. (2023)": formatting
Thanks, I have changed \citet to \citep.

L. 178: "in contrast": I was a bit confused about where the contrast was. There could be a sentence: "WD frequency predominantly relates to jet strength, whereas WD intensity predominantly relates to jet position, (although both jet statistics remain significantly correlate with each of the WD statistics)"
This is a much clearer statement. I have included it in the revision (replacing the original).

Also, the reasons for the difference in correlation could be clarified more explicitly: A stronger jet either meanders more or simply advect more quickly the meanders (actually, which is it?), resulting in a higher number of meanders (and thus WD) at a

specific location. (This is a large-scale process)

By contrast, the precise positioning of the jet is really what enables the meanders to grow into powerful WDs, through the relief interaction. (it is a more local scale process)

I agree that this is worth including and have done so. Out of interest, the increase in jet speed (~6%) cannot explain the increase in WD frequency (~13%) for January over the study period through faster advection alone. The revised text reads: "This arises from both large-scale and local processes. On the synoptic scale, a stronger jet meanders more, resulting in a higher number of WDs at a specific location. In contrast, on the local scale, the precise positioning of the jet is what enables the meanders to grow into powerful WDs, through interaction with the orography.".

L. 182: "the sharp increase in jet strength": the jet latitude has actually a slightly higher absolute correlation. In any case, both jet strength and intensity seem important.

Yes, but for April (to which this refers) only the trend in jet strength is significant. The trend in jet latitude is insignificant.

L. 183: "We can probably rule out jet strength as a confounding factor": Should we? The correlation is significant, why not try to understand why? Could it be that the jet is more wavy, and therefore the temporal averaging makes the jet appear weak?

You're right, this is worth exploring further. Below I plot trends in two simple metrics for jet waviness, computed using daily data, for April (blue) and June (red). "jet instability" is the standard deviation of $v$ within 10° north or south of the jet axis. "jet waviness" is the standard deviation of the jet axis latitude between 50 and 80°E.

[Figure]

[Figure]

Neither metric shows a significant trend in June (to which the original statement refers), and so my original stated hypothesis is probably correct. I state this in the revised manuscript: "We can probably rule out jet strength as a confounding factor in these months because the only known mechanism through which a weaker jet produces more WDs is through a latitudinal shift in its position; otherwise decreased baroclinicity would cause a decline in frequency. Nor is it a case that temporal averaging of a wavier jet makes it appear more weak -- there is no

significant positive trend in metrics of jet waviness in these transition months (not shown).".

L. 185: "between these two effects." which effects? The effect of jet latitude is the same between April and June, and the effect of jet strength is indeed the opposite between the two months but discarded...
This refers to the relative effects of jet strength and jet latitude, which I will clarify in the revised manuscript. The effect of jet latitude in April and June is not the same because jet latitude does not have a significant trend in April, whereas it does in June. The opposite (and perhaps counterintuitive) effect of jet strength in June is discussed as a "confounding factor" both in the manuscript and in my response to your previous comment.

L. 201: "are much stronger in winter months": This is not a result of this study. Hunt et al, 2018a already established it.
I have clarified that this result has already been found in an earlier study: "As previously found in Hunt et al. (2018a), WDs, as measured by their..."

L. 228: "There is has been" -> There has been
Will fix.